# Histologic-Based Tumor-Associated Immune Cells Status in Clear Cell Renal Cell Carcinoma Correlates with Gene Signatures Related to Cancer Immunity and Clinical Outcomes

**DOI:** 10.3390/biomedicines10020323

**Published:** 2022-01-29

**Authors:** Chisato Ohe, Takashi Yoshida, Junichi Ikeda, Toyonori Tsuzuki, Riuko Ohashi, Haruyuki Ohsugi, Naho Atsumi, Ryosuke Yamaka, Ryoichi Saito, Yoshiki Yasukochi, Koichiro Higasa, Hidefumi Kinoshita, Koji Tsuta

**Affiliations:** 1Department of Pathology, Kansai Medical University, Hirakata 573-1191, Japan; ikedaj1985@gmail.com (J.I.); naatsumi@hirakata.kmu.ac.jp (N.A.); yamakary@hirakata.kmu.ac.jp (R.Y.); tsutakoj@hirakata.kmu.ac.jp (K.T.); 2Department of Urology and Andrology, Kansai Medical University, Hirakata 573-1191, Japan; yoshidtk@takii.kmu.ac.jp (T.Y.); haosugi@gmail.com (H.O.); saitor@hirakata.kmu.ac.jp (R.S.); kinoshih@hirakata.kmu.ac.jp (H.K.); 3Department of Surgical Pathology, Aichi Medical University Hospital, Nagakute 480-1195, Japan; tsuzuki@aichi-med-u.ac.jp; 4Histopathology Core Facility, Faculty of Medicine, Niigata University, Niigata 951-8510, Japan; riuko@med.niigata-u.ac.jp; 5Department of Genome Analysis, Institute of Biomedical Science, Kansai Medical University, Hirakata 573-1191, Japan; yasukocy@hirakata.kmu.ac.jp (Y.Y.); higasako@hirakata.kmu.ac.jp (K.H.)

**Keywords:** clear cell renal cell carcinoma, immunophenotype, immunohistochemistry, gene expression signatures, cancer immunity, clinical outcome

## Abstract

The three-tier immunophenotype (desert, excluded, and inflamed) and the four-tier immunophenotype (cold, immunosuppressed, excluded, and hot) have been linked to prognosis and immunotherapy response. This study aims to evaluate whether immunophenotypes of clear cell renal cell carcinoma, identified on hematoxylin and eosin-stained slides, correlate with gene expression signatures related to cancer immunity, and clinical outcomes. We evaluated tumor-associated immune cells (TAICs) status using three methodologies: three-tier immunophenotype based on the location of TAICs, four-tier immunophenotype considering both the location and degree of TAICs and inflammation score focusing only on the degree of TAICs, using a localized clear cell renal cell carcinoma cohort (*n* = 436) and The Cancer Genome Atlas (TCGA)-KIRC cohort (*n* = 162). We evaluated the association of the TAICs status assessed by three methodologies with CD8 and PD-L1 immunohistochemistry and immune gene expression signatures by TCGA RNA-sequencing data. All three methodologies correlated with immunohistochemical and immune gene expression signatures. The inflammation score and the four-tier immunophenotype showed similarly higher accuracy in predicting recurrence-free survival and overall survival compared to the three-tier immunophenotype. In conclusion, a simple histologic assessment of TIACs may predict clinical outcomes and immunotherapy responses.

## 1. Introduction

Recently, the prognosis of metastatic renal cell carcinoma (RCC) has improved by the efficacy of immune checkpoint inhibitors (ICIs) such as agents directed against cytotoxic T-lymphocyte antigen 4 (CTLA-4), programmed death-1 (PD-1), and programmed death ligand-1 (PD-L1) [1] in addition to the anti-vascular endothelial growth factor (VEGR) targeted therapy such as tyrosine kinase inhibitors (TKIs) [2]. Comprehensive genomic investigations have provided a biology-based tumor immune microenvironment for treatment selection using genomic and transcriptomic analysis [3,4,5]. Several clinical trials on clear cell RCC (ccRCC) to evaluate their potential predictive value of TKIs, ICIs alone, or in combination, for patients with metastatic RCC revealed the association of gene expression signatures related to angiogenesis, effector T-cell, and myeloid inflammation with clinical outcome [6,7].

Since multiregional molecular genetic studies have largely been performed without consideration of morphology, the investigation regarding the correlation between histologic immune status and underlying genomes has been limited [8]. Genetic analysis and multiplex immunofluorescence/immunohistochemical staining remain challenging in routine clinical practice. Thus, a simple histologic-based assessment of the tumor immune microenvironment is urgently needed. A standardized methodology to assess tumor-infiltrating lymphocytes (TILs) has been proposed in solid tumors to assess the immune response to tumors [9]. On the other hand, recent evidence suggests that tumor immunity has been categorized into three immunophenotypes based on the location of TILs as desert, non-inflamed; excluded, peritumoral immune infiltration; and inflamed, intratumoral immune infiltration [10,11]. However, the prognostic and predictive value of TILs for RCCs remains under investigation.

Regarding the histological features of ccRCC, although the most common architectural pattern was compact small nests with an extensive vascular network, various architectural patterns have been reported [12,13]. Our research group has recently established a vascularity-based architectural classification for ccRCC, which significantly correlated with angiogenesis and immune gene signatures [14]. Although we also confirmed that the immunophenotype, such as desert, excluded, and inflamed, was highly associated with the vascularity-based architectural classification, the degree of inflammation was variable in both the excluded and inflamed types. Recent advances of immune contexture, including the novel concept of a combination of immune variables, such as the nature, density, immune functional orientation, and distribution of immune cells within the tumor, has led to four classifications (cold, immunosuppressed, excluded, and hot) [15]. At present, there is little evidence regarding which histologic-based assessment of immunophenotype for ccRCC correlates with the underlying mechanism of cancer immunity.

The current study aims to evaluate whether immunophenotypes of ccRCC, identified on hematoxylin and eosin (H&E)-stained slides, correlated with the gene expression signature related to cancer immunity, and clinical outcomes. In the present study, we first validated the histologic-based tumor-associated immune cells (TAICs) status using three methodologies: three-tier immunophenotype based on the location of TAICs, four-tier immunophenotype considering both the location and degree of TAICs, and inflammation score focusing only on the TAIC degree. Then, we correlated findings with CD8 and PD-L1 immunohistochemical expression and gene expression signatures related to cancer immunity by RNA-sequencing data available from the Cancer Genome Atlas (TCGA) [16,17]. Furthermore, we compared the correlation and prognostic accuracy among three methods and evaluated the association of histologic-based TAICs with pathological prognostic factors. 

## 2. Materials and Methods

### 2.1. Data Collection

In the present study, we used the same two cohorts as previously reported [14]; 436 cases with localized ccRCC (cT1-4N0-1M0) as the principal cohort and 162 ccRCC cases from the TCGA-KIRC cohort. Cases with presurgical treatment with TKIs or ICIs were not included in the principal cohort. The study of the principal cohort was approved by the Institutional Review Board (No. 2018109 (28/11/2018) and No. 2020222 (09/12/2020)) per the Declaration of Helsinki. TCGA whole slide images were accessed via the Cancer Digital Slide Archive [18]. We retrieved information regarding the tumor stage assessed by the 2017 TNM staging system [19], World Health Organization (WHO)/International Society of Urological Pathology (ISUP) nucleolar grade, and pathological prognostic factors, such as sarcomatoid/rhabdoid components and tumor-specific necrosis [20], from our institutional ccRCC database as previously described [14,21,22]. 

The study design is shown in Figure 1.

### 2.2. Histological Evaluation of Tumor Immune Microenvironment 

To assess the tumor immune microenvironment, TAICs including both mononuclear cells and granulocytes were evaluated using whole H&E-stained slides. One representative slide including the highest-grade area (an average of five slides containing tumors per case) and one representative whole slide image was assessed in the principal cohort and TCGA cohort, respectively. The TAICs status was evaluated using three methods (shown in Table 1) by a genitourinary pathologist (C.O.) blinded to clinical outcomes. The three-tier immunophenotype was categorized based on the location of TAICs regardless of the degree of TAICs as follows: desert, no TAICs; excluded, peritumoral TAICs; and inflamed, intratumoral TAICs, as previously described [14]. The four-tier immunophenotype was categorized considering both the location and degree of TAICs as follows: cold, no TAICs; immunosuppressed, focal or low TAICs, regardless of the TAICs location; excluded, diffuse or high peritumoral TAICs; and hot, diffuse or high intratumoral TAICs, as previously described [15]. Inflammation score was categorized focusing only on the degree of TAICs, regardless of the location, as follows: score 0, no TAICs; score 1, focal or low TAICs; and score 2, diffuse or high TAICs. Figure 2 shows representative images of the categorization of TAICs status.

### 2.3. Immunohistochemical Analysis

We used the previously reported data assessed on tissue microarray sections from 2 mm-cores of formalin-fixed, paraffin-embedded (FFPE) blocks to evaluate the association of the histologic-based TAICs with CD8 and PD-L1 immunohistochemical expression for the principal cohort (*n* = 121 and 429, respectively) [21,22,23]. We performed immunohistochemistry (IHC) using Leica Bond-III (Leica Biosystems, Melbourne, Australia) and a Ventana Discovery Ultra autostainer (Roche Diagnostics K.K, Tokyo, Japan). Primary antibodies against CD8 (4B11, Prediluted; Leica Biosystems, Newcastle Upon Tyne, UK) and PD-L1 (28-8, 1:400; Abcam, Cambridge, MA, USA) were used to visualize with a BOND Polymer Refine Detection (Leica Biosystems) and OptiView DAB IHC Detection Kit (Ventana Medical System, Tucson, AZ, USA), respectively. We evaluated the density of CD8+ TILs (calculated as the number of cells/mm^2^ per cores) and we semiquantitatively assessed the membranous staining pattern of PD-L1 in tumor cells using the H-score (0–300) as previously described [22,23]. 

### 2.4. Gene Expression Analysis

To validate the correlation between the histologic-based TAICs and gene expression signatures related to cancer immunity, the IMmotion 150 gene signatures [7], consisting of angiogenesis, immune and antigen presentation, and myeloid inflammation, were extracted from the RNA-sequencing data of TCGA according to previous reports [14,23]. Three gene signatures related to the underlying mechanisms of ICIs response were defined as follows: effector T-cell: *CD8A, IFNG, GZMA, GZMB, PRF1,* and *EOMES*; immune checkpoint: *CD274 (PD-L1), CTLA4,* and *TIGIT*; and myeloid: *CXCL1, CXCL2, CXCL3, IL6*, and *PTGS2*, as previously described [14]. To calculate gene signature scores, each gene score was normalized by the z-score across all patients and averaged to create signature scores for each patient [24]. The TCGA RNA-sequencing data were downloaded as described previously [25]. 

### 2.5. Statistical Analysis

All statistical analyses were performed using EZR version 1.54 (Saitama Medical Center, Jichi, Japan) [26]. All continuous data were shown as median-valued and interquartile ranges (IQR). A Chi-squared test for categorical variables and one-way ANOVA analysis, Mann–Whitney U test, or Kruskal–Wallis test for non-parametric variables were used to evaluate the statistical significance among three or four groups. The F-statistic in the linear regression analysis was calculated to determine the statistical significance among the three. The study outcome measure was recurrence-free survival (RFS), defined as the time from surgery to initial local or distant metastasis shown on imaging, and overall survival (OS) was defined as the time from surgery to any cause of death in the principal cohort and TCGA cohort, respectively. RFS or OS was assessed by the Kaplan–Meier method with the log-rank and the Cox proportional hazards models. Harrell’s concordance index (c-index) was used to compare the predictive accuracy of the Cox models. A two-sided *p* < 0.05 was considered statistically significant.

## 3. Results

### 3.1. Clinicopathological Characteristics

In the principal cohort, the median age at ccRCC diagnosis was 65 years (IQR, 56–73 years). In total, 103 (23.7%) of the tumors were in the high stage (TNM stage III or IV) and 142 (32.5%) were of a high WHO/ISUP grade (3 or 4). Of the 436 patients, 57 (13.1%) experienced a recurrence and 15 (3.4%) died of ccRCC during a median follow-up period of 61.7 months (IQR, 33.8–93.6), as previously reported [14]. 

In the TCGA cohort, of the 162 patients with ccRCC, 67 (41.4%) died during a median follow-up period of 1173 days (IQR, 563–1779). Of the tumors, 65 (40.1%) were in the high stage (TNM stage III or IV) and 69 (42.5%) were a high WHO/ISUP grade (3 or 4). The histologically identified TAICs status assessed by the three methods is summarized in Table 2. A statistically significant relationship was found between two cohorts in each methodology. The clinicopathological characteristics of three-tier and four-tier immunophenotype and inflammation scores in both the principal and TCGA cohorts are shown in Appendix A.

### 3.2. Comparison of Immunohistochemical Expression among Three-Tier and Four-Tier Immunophenotype and Inflammation Score

CD8+ TILs were significantly enriched in the inflamed compared to the desert (*p* < 0.001), followed by the excluded in the three-tier immunophenotype. CD8+ TILs were significantly distributed among all four types in the four-tier immunophenotype, whereas scores 1 and 2 made up a significantly higher density of CD8+ TILs compared to score 0 in the inflammation score (*p* < 0.01) (Figure 3A). Representative examples of CD8 IHC are shown in Figure 3B. Inflamed and excluded showed significantly higher PD-L1 expression compared to the desert in the three-tier immunophenotype (*p* < 0.01). In the four-tier immunophenotype, PD-L1 expression of hot and excluded was significantly higher compared to immunosuppressed and cold (*p* < 0.01). In the inflammation score, scores 2 and 1 had significantly higher expression compared to score 0 (*p* < 0.01) (Figure 3C). Representative examples of PD-L1 IHC are shown in Figure 3D.

### 3.3. Comparison of Gene Expression Signatures among Three-Tier and Four-Tier Immunophenotypes and Inflammation Scores

In the TCGA cohort, significant difference among the three scores were found in effector T-cell and myeloid gene signatures in the three-tier immunophenotype (both *p* < 0.05), whereas no significant difference was found in the immune checkpoint gene signature (Figure 4A). In the four-tier immunophenotype, effector T-cell and immune checkpoint gene signature was significantly enriched in hot and excluded compared to immunosuppressed and cold (*p* < 0.001). Furthermore, significant differences among the four types were found in the myeloid gene signature, with the highest expression of excluded (*p* < 0.05) (Figure 4B). In the inflammation score, the effector T-cell and immune checkpoint gene signature were significantly enriched in score 2 (*p* < 0.001), followed by score 1 (*p* < 0.001), and significant differences were noted among the three types in myeloid gene signature (*p* < 0.01) (Figure 4C). Correlation analysis among the three methodologies showed that the inflammation score was the highest in the effecter T-cell (f-statistic = 23.92, *p* < 0.001), immune checkpoint (f-statistic = 26.73, *p* < 0.001), and myeloid (f-statistic = 7.23, *p* = 0.008) gene signatures. Comparing the three-tier and four-tier immunophenotypes, the four-tier immunophenotype was higher in all gene signatures: effecter T-cell (f-statistic = 14.29, *p* < 0.001), immune checkpoint (f-statistic = 15.91, *p* < 0.001), and myeloid (f-statistic = 6.77, *p* < 0.001)), compared to the three-tier immunophenotypes (effecter T-cell (f-statistic = 3.48, *p* = 0.03), immune checkpoint (f-statistic = 4.08, *p* = 0.02), and myeloid (f-statistic = 1.59, *p* = 0.21) (Figure 5).

### 3.4. Comparison of Patient Outcome among Three-Tier and Four-Tier Immunophenotype and Inflammation Score

The Kaplan–Meier survival analysis showed a five-year RFS rate of 78.7% for inflamed and 74.5% for excluded versus 95% for desert in the three-tier immunophenotype; 79.8% for immunosuppressed, 78.6% for hot, and 67.3% for excluded versus 95% for cold in the four-tier immunophenotype; and 79.8% for score 1 and 71.7% for score 2 versus 95% for score 0 in the inflammation score (Figure 6A–C). The Kaplan–Meier survival analysis of the TCGA-cohort showed a similar trend compared to the principal cohort: a five-year OS rate of 45.9% for inflamed and 31.0% for excluded versus 74.9% for desert in the three-tier immunophenotype; 44.9% for immunosuppressed and 28.6% for hot versus 74.9% for cold in the four-tier immunophenotype (excluded could not be assessed due to the limited number of cases); and 44.9% for score 1 and 20.8% for score 2 versus 74.9% for score 0 in the inflammation score (Figure 6D–F). Moreover, the four-tier immunophenotype and inflammation score more accurately predicted RFS or OS than the three-tier immunophenotype in both cohorts (c-index = 0.692 vs. 0.672 for the principal cohort and 0.685 and 0.684 vs. 0.661 for the TCGA cohort) (Figure 7A,B). Univariate associations with recurrence after nephrectomy in patients with clear cell renal cell carcinoma are shown in Table 3.

### 3.5. Association of Inflammation Score with Pathological Prognostic Factors

Considering both correlation analysis and the prognostic accuracy among the three methodologies, the inflammation score was the most correlated with the gene expression signatures (Figure 5) and prognostic prediction (Figure 7). Regarding the association of the inflammation score with TNM stage pathological prognostic factors, WHO/ISUP grade, sarcomatoid/rhabdoid components, and necrosis factors, significant differences among scores were found in both the principal and TCGA cohort (all *p* < 0.001; Figure 8A,B).

## 4. Discussion

Using the localized ccRCC and TCGA-KIRC cohorts, we demonstrated an association between the three methodologies of histologic-based TAICs considering the location and/or degree of TAICs with immunohistochemical expression and gene expression signatures related to cancer immunity. Although all three methodologies correlated with immunohistochemical and immune gene expression signatures, we showed the inflammation score based on the degree of TAICs the most correlated with effector T-cell, immune checkpoint, and myeloid gene signatures among the three methodologies. Regarding the prognostic prediction, inflammation score and four-tier immunophenotype based on both the location and degree of TAICs showed similarly higher accuracy in predicting RFS or OS compared to the three-tier immunophenotype based on the TAICs location. Thus, a simple histologic assessment of TIACs may predict clinical outcomes and immunotherapy responses.

While the previously reported methodology focused on the TILs, we assessed the TAICs status including both mononuclear cells and granulocytes on only H&E-stained slides. Currently, the PD-L1 expression on immune or tumor cells, the extensive infiltration of CD8+ TILs, and the high tumor mutational burden are the most sensitive and specific biomarkers of clinical response to checkpoint blockade in several solid tumors [10]. Although many different approaches can assess and describe the immune response, a standardized methodology to assess TILs on H&E-stained sections for solid tumors is a valid and reproducible biomarker in routine clinical practice [9,27,28]. However, whether the assessment of only TILs reflects cancer immunity remains unclear.

Among various features of the tumor microenvironment of ccRCC, Cai and Christie et al. have recently shown that the intratumoral neutrophilic infiltration of ccRCC is one independent predictor for disease-free survival [13]. Although effective immunotherapy promotes the killing of cancer cells by cytotoxic T cells [10], CD4+ helper T cells, regulatory T cells (Tregs), effector T-cells, macrophages, and neutrophils are also important immune factors that can either promote or block tumor development [29]. Because host immunity, consisting of various types of TAICs and cellular matrix, plays an active role in controlling tumor growth and metastatic spreading [30,31], our simple and comprehensive assessment of TAICs is reasonable in this current study.

To the best of our knowledge, histologic-based TAICs status for ccRCC has not been extensively investigated. We first compared the association of the three TAICs status with immunohistochemical and gene expression and oncological outcomes, using three methodologies: the three-tier immunophenotypes inflamed, immune desert, or immune-excluded based on the spatial localization of immune cells [14,32]; the four-tier immunophenotypes referred to the novel concept of hot, excluded, immunosuppressed, and cold phenotypes, including the nature, density, immune functional orientation, and distribution of immune cell parameters [14]; and our proposed inflammation score focuses only on the degree of TAICs. Although these phenotypes were originally defined based on CD8+ TIL, we categorized these phenotypes by histologically identified TAICs. Nevertheless, TAICs detected on H&E-stained slides are highly correlated with immunohistochemical expression of CD8 and effecter T-cell gene signatures, including *CD8A*, which suggest histologically identified TAICs within ccRCC predominantly include CD8+ TILs.

We found that the four-tier immunophenotype considering both the location and degree of TAICs was more highly correlated with gene expression signatures related to cancer immunity compared to the three-tier immunophenotype based on the location of TAICs. In the present study, however, we demonstrated that the inflammation score considering only the degree of TAICs was the most correlated with both gene expression signatures and prognostic prediction among the three methodologies. Accordingly, our proposed simple inflammation score of TAICs by H&E-stained slides might be easier to apply than other methods in a routine clinical setting.

Concerning the prognostic prediction of infiltrating immune cells, evidence suggests that increased infiltrations with CD8+ and Th1-CD4+ T-cells are associated with a good prognosis in most solid tumors [33]. Our prior study of bladder cancer revealed that high TAICs were significantly associated with a favorable oncological outcome compared to low TAICs [34], which is supported by other studies [35]. Contrary to other solid tumors, RCC enriched in high-density CD8+ TILs are associated with poor prognosis [36,37,38]. Consist with previous studies of RCC, we found that degree of inflammation could stratify patient outcome, which implicated increased TAICs involvement in the formation of pre-metastatic niches.

Several studies have revealed that patients with the immune-excluded phenotype had a poor prognosis compared to patients with the inflamed or desert phenotype in various tumors because immune-excluded tumors were immunosuppressed by T-cells embedded in the tumor stromal microenvironment with upregulation of TGFβ, myeloid inflammation, and angiogenesis [10,14,39]. Consistent with previous studies, the prognosis of the excluded type was the worst in both the three-tier and four-tier immunophenotypes. Interestingly, our results showed that the myeloid gene signature of the excluded type was significantly high in the four-tier immunophenotype, although myeloid gene signatures were not statistically significant in the three-tier immunophenotype. Thus, when considering the myeloid-derived suppressor cells (MDSCs) involved in RCC progression and drug resistance, the four-tier immunophenotype would help determine patient selection for immunotherapy. Further studies are needed to determine which TAICs assessment methodology plays a predictive role in the selection of ICIs for ccRCC.

Regarding the association of pathological prognostic factors, we found that a dedifferentiation form of ccRCC, such as sarcomatoid/rhabdoid, was significantly correlated with inflammation score 3, followed by score 2, which corresponded with the enrichment of immune checkpoint signatures. Beuselinck et al. also demonstrated a correlation between poor tumor cell differentiation and overexpression of immunomodulatory molecules in ccRCCs using gene expression analysis [40]. These results corroborate that sarcomatoid/rhabdoid features could be potential surrogate markers of ICI response [41].

Our study has several limitations. First, this was a retrospective study including the excluded four-tier immunophenotype in the TCGA cohort with a small sample size. Because this study cohort was the same as that which was previously described [14], multiple institutional cohorts should be validated. Second, CD8 and PD-L1 immunohistochemical expression was evaluated with TMAs constructed with two representative cores and not using whole sections. Third, we correlated TAICs assessment and gene expression signatures using only the TCGA cohort. Fourth, the principal cohort candidates were localized ccRCC patients with only an endpoint of RFS. The candidate for the TCGA-KIRC cohort included metastatic ccRCC patients with only an available endpoint of OS, as previously described [14]. Although there was a statistically significant relationship between two cohorts in each methodology (Table 2), we have confirmed a similar trend in the association of TAICs status with clinicopathological factors (Appendix A). Despite these limitations, we comprehensively showed the association of histologic-based TAICs with gene signatures related to cancer immunity and clinical outcomes.

## 5. Conclusions

Our current work showed histologic-based TAICs status identified by H&E-stained slides correlated with effector T-cell, immune checkpoint, myeloid gene signatures, and clinical outcomes. Histologic assessment of TIACs may serve as a surrogate for gene expression signatures related to immunotherapy response and support prognostic prediction. Our approach using only H&E-stained slides can be performed in routine clinical practice at a low cost.

## Figures and Tables

**Figure 1 biomedicines-10-00323-f001:**
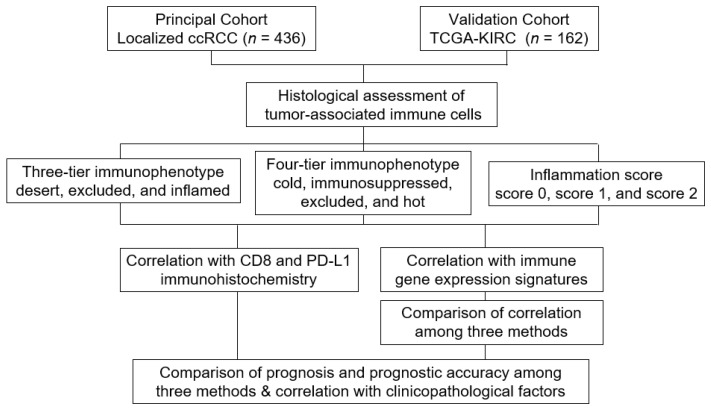
Study design.

**Figure 2 biomedicines-10-00323-f002:**
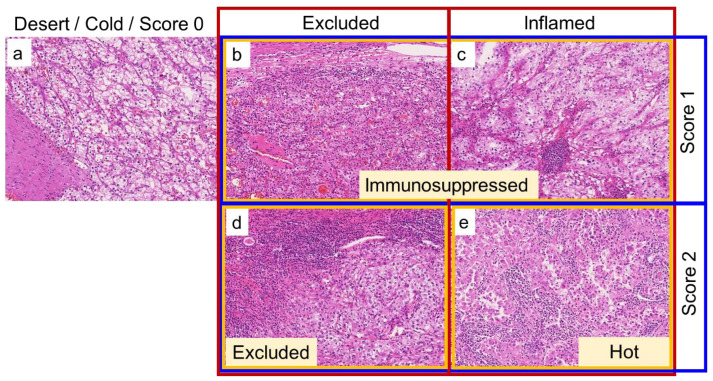
Representative images of how to categorize the tumor-associated immune cells (TAICs) status on hematoxylin and eosin (H&E)-stained slides. Red, yellow, and blue boxes represent the three-tier immunophenotype, four-tier immunophenotype, and inflammation score, respectively. (**a**): No TAICs, desert (three-tier immunophenotype), cold (four-tier immunophenotype), and score 0 (inflammation score). (**b**): Focal or low peritumoral TAICs, excluded (three-tier immunophenotype), immunosuppressed (four-tier immunophenotype), and score 1 (inflammation score). (**c**): Focal or low intratumoral TAICs, inflamed (three-tier immunophenotype), immunosuppressed (four-tier immunophenotype), and score 1 (inflammation score). (**d**): Diffuse or strong peritumoral TAICs, excluded (three-tier and four-tier immunophenotype), and score 2 (inflammation score). (**e**): Diffuse or strong intratumoral TAICs, inflamed (three-tier immunophenotype), hot (four-tier immunophenotype), and score 2 (inflammation score). All images are taken at 10× magnification.

**Figure 3 biomedicines-10-00323-f003:**
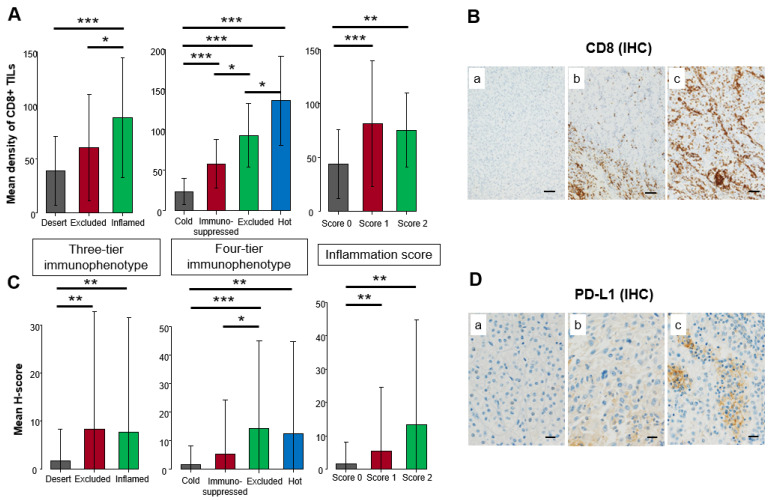
Comparison of immunohistochemical expression among three-tier and four-tier immunophenotype and inflammation score in the principal cohort. (**A**) Mean density of CD8+ tumor-infiltrating lymphocytes (TILs). (**B**) Representative example of CD8 immunohistochemistry (IHC). The brown color showed positive staining of immunohistochemistry whereas the blue color showed a counterstain of cell nuclei. (**a**). none, (**b**). CD8+ TILs accumulated in the peritumoral area, (**c**). CD8+ TILs infiltrated in the intratumoral area. Scale bar: 200 µm. (**C**) Mean H-score of PD-L1 expression on cancer cells. (**D**) Representative example of PD-L1 IHC. The brown color showed positive staining of immunohistochemistry whereas the blue color showed a counterstain of cell nuclei. (**a**). none, (**b**). focal weak positivity, (**c**). focal strong positivity. Scale bar: 20 µm. One-way analysis was used for statistical analysis (* *p* < 0.05, ** *p* < 0.01, *** *p* < 0.001 assessed by Mann–Whitney U-test).

**Figure 4 biomedicines-10-00323-f004:**
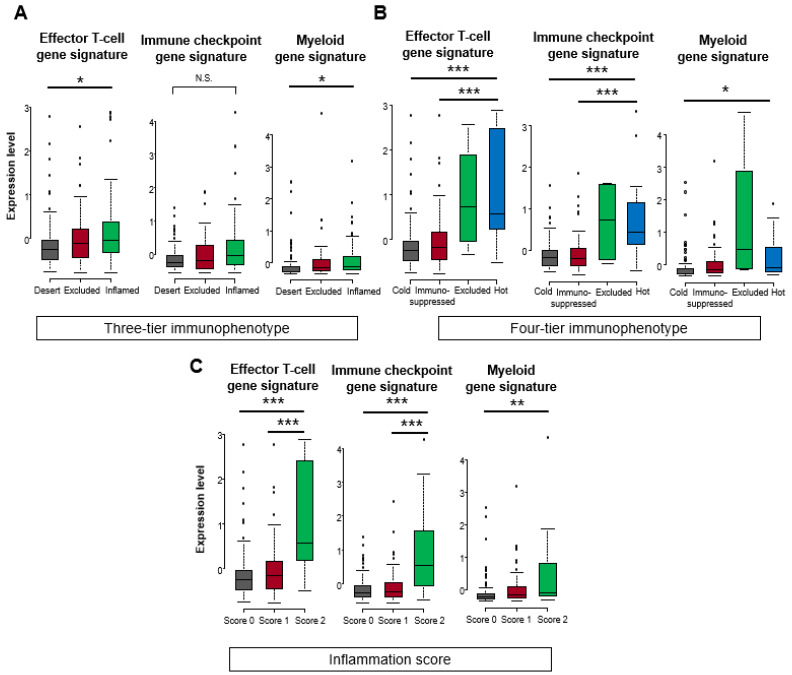
Comparison of gene expression signatures among three-tier and four-tier immunophenotype and inflammation score in the TCGA cohort. Comparison scores (mean Z-score) related to effector T-cell, immune checkpoint, and myeloid among (**A**) three-tier immunophenotype, (**B**) four-tier immunophenotype, and (**C**) inflammation score. Kruskal–Wallis test was used for statistical analysis (N.S. means not statistically significant. * *p* < 0.05, ** *p* < 0.01, *** *p* < 0.001 assessed by Mann–Whitney U-test).

**Figure 5 biomedicines-10-00323-f005:**
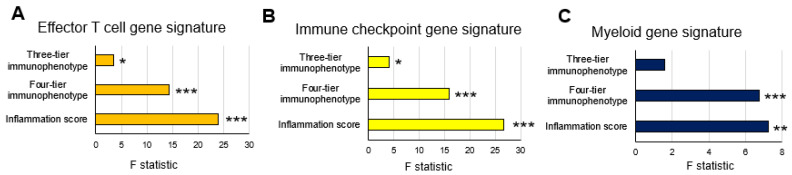
Correlation analysis between three evaluation methods of the tumor immune microenvironment and gene expression signatures in the TCGA cohort: (**A**) effector T-cell gene signature; (**B**) immune checkpoint gene signature; and (**C**) myeloid gene signature. * *p* < 0.05, ** *p* < 0.01, *** *p* < 0.001 using linear regression analysis.

**Figure 6 biomedicines-10-00323-f006:**
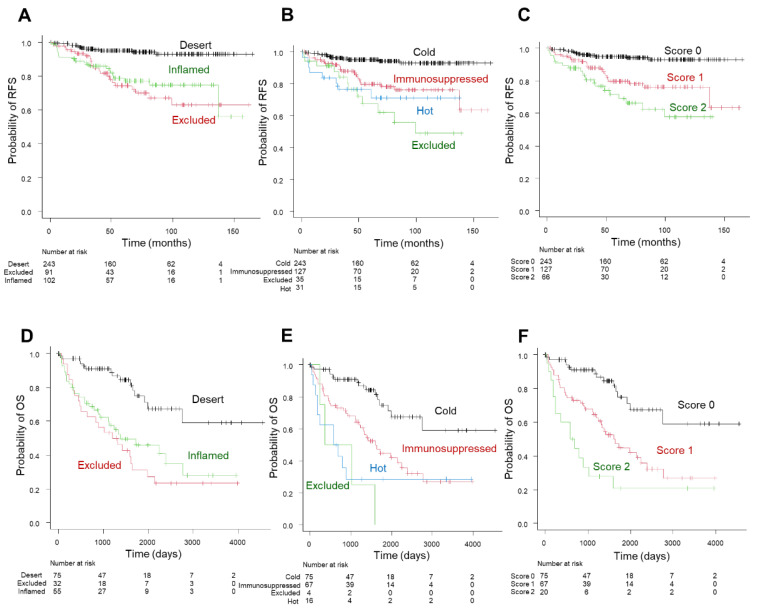
Prognostic significance of the three-tier and four-tier immunophenotype and inflammation score in the principal cohort (**A**–**C**) and TCGA cohort (**D**–**F**). Kaplan–Meier curve of recurrence-free survival (RFS) in the three-tier immunophenotype (**A**); four-tier immunophenotype (**B**); and inflammation score (**C**). Kaplan–Meier curve of overall survival (OS) in three-tier immunophenotype (**D**); four-tier immunophenotype (**E**); and inflammation score (**F**).

**Figure 7 biomedicines-10-00323-f007:**
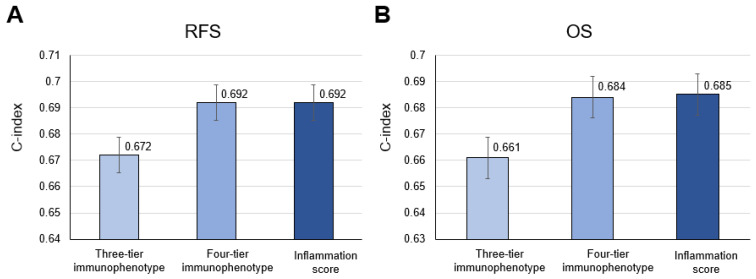
Comparison of accuracy among the three-tier and four-tier immunophenotype and inflammation score in (**A**) recurrence-free survival (RFS) and (**B**) overall survival (OS) in the principal cohort and the TCGA cohort, respectively.

**Figure 8 biomedicines-10-00323-f008:**
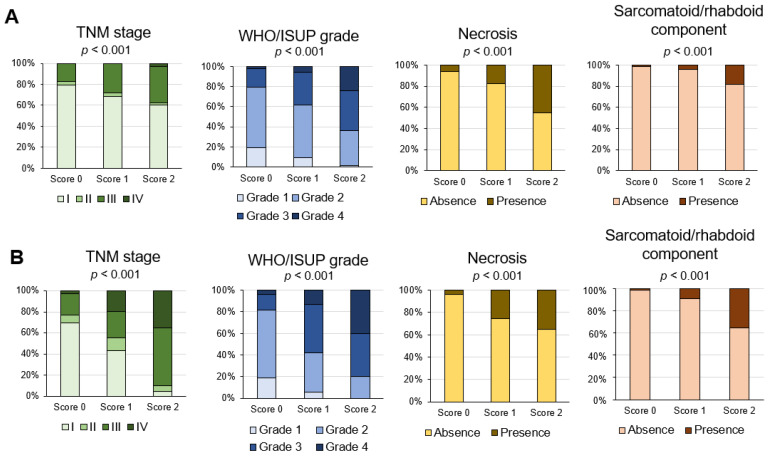
Association of inflammation score with pathological prognostic factors in the principal cohort (**A**) and the TCGA-KIRC cohort (**B**). Percentage of cases of each inflammation score and pathological prognostic factor.

**Table 1 biomedicines-10-00323-t001:** Comparison of three evaluation methods of the tumor immune microenvironment.

Three-tier immunophenotype	Based on the location of TAICs, regardless of the TAICs degree			
	Desert	Excluded	Inflamed	
	No TAICs	Peritumoral TAICs	Intratumoral TAICs	
Four-tier immunophenotype	Considering both the location and degree of TAICs			
	Cold	Immunosuppressed	Excluded	Hot
	No TAICs	Focal or low TAICs, regardless of the TAICs location	Diffuse or high peritumoral TAICs	Diffuse or high intratumoral TAICs
Inflammation score	Based on the degree of TAICs, regardless of the TAICs location			
	Score 0	Score 1	Score 2	
	No TAICs	Focal or low TAICs	Diffuse or high TAICs	

TAICs, tumor-associated immune cells.

**Table 2 biomedicines-10-00323-t002:** Tumor-associated immune cells status assessed by three methods.

Variables	Principal Cohort *n* = 436	TCGA Cohort *n* = 162	*p* Value
Three-tier immunophenotype, *n* (%)			0.029
Desert	243 (55.7)	75 (46.3)	
Excluded	91 (20.9)	32 (19.8)	
Inflamed	102 (23.4)	55 (34.0)	
Four-tier immunophenotype, *n* (%)			0.003
Cold	243 (55.7)	75 (46.3)	
Immunosuppressed	127 (29.1)	67 (41.4)	
Excluded	35 (8.0)	4 (2.5)	
Hot	31 (7.1)	16 (9.9)	
Inflammation score, *n* (%)			0.018
Score 0	243 (55.7)	75 (46.3)	
Score 1	127 (29.1)	67 (41.4)	
Score 2	66 (15.1)	20 (12.3)	

**Table 3 biomedicines-10-00323-t003:** Univariate associations with recurrence after nephrectomy in patients with clear cell renal cell carcinoma.

	Cohort	Principal Cohort		TCGA Cohort	
	Variables	HR (95% CI)	*p* Value	HR (95% CI)	*p* Value
Three-tier immunophenotype	Desert	1 (ref.)		1 (ref.)	
	Excluded	5.23 (2.63–10.39)	<0.001	3.93 (2.05–7.54)	<0.001
	Inflamed	4.44 (2.23–8.81)	<0.001	3.10 (1.66–5.79)	<0.001
Four-tier immunophenotype	Cold	1 (ref.)		1 (ref.)	
	Immunosuppressed	3.87 (1.97–7.59)	<0.001	2.94 (1.62–5.36)	<0.001
	Excluded	7.60 (3.47–16.7)	<0.001	8.49 (2.79–25.9)	<0.001
	Hot	5.82 (2.41–14.1)	<0.001	5.20 (2.38–11.4)	<0.001
Inflammation score	Score 0	1 (ref.)		1 (ref.)	
	Score 1	3.87 (1.97–7.59)	<0.001	2.94 (1.62–5.36)	<0.001
	Score 2	6.77 (3.37–13.6)	<0.001	5.79 (2.82–11.9)	<0.001

HR, hazard ratio; CI, confidence interval; ref., reference.

## Data Availability

The data are available upon reasonable request by contacting the corresponding author.

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
