# Peer review of "Histologic-Based Tumor-Associated Immune Cells Status in Clear Cell Renal Cell Carcinoma Correlates with Gene Signatures Related to Cancer Immunity and Clinical Outcomes"

_biomedicines, 2022, doi:10.3390/biomedicines10020323_

Round 1
Reviewer 1 Report
This paper evaluates the association of the histologic-based tumor-associated immune cells status with gene signatures related to cancer immunity and clinical outcomes. The authors conducted one of the most comprehensive studies in this field focusing on RCC and analysis two different cohort. While the paper is interesting for basic science and understanding of immunoregulation, the potential clinical translation remains uncertain.
Major:
1. Introduction/Discussion: Please elaborate on the clinical rational for this study. Gene signatures are known to be prognostic. However, they do not guide clinical decision making and especially do not change the selection of therapy.
2. Can you provide data showing the correlation between immune cell status and clinical risk stratification based on IMDC Risk Criteria? Most studies do not improve the clinical standard of IMDC risk criteria in RCC…
3. Table 2: Please add p-values
Minor:
Paragraph 2 of the Introduction: Please improve the clinical rational to perform the study. In addition to “technological and economical issues” the lack of a clear target makes most approaches a failure. Therefore, most genomic applications have failed in routine testing in urology/RCC despite being feasible (and even reimbursed in some countries) (e. g. Rodler et. al., European Journal of Cancer 2021). Consequently, an easy to perform histological assessment to predict response would be helpful. Please discuss…
Author Response
Response to Reviewer 1 CommentsThis paper evaluates the association of the histologic-based tumor-associated immune
cells status with gene signatures related to cancer immunity and clinical outcomes. The
authors conducted one of the most comprehensive studies in this field focusing on
RCC and analysis two different cohort. While the paper is interesting for basic science
and understanding of immunoregulation, the potential clinical translation remains
uncertain.
Thank you for your review. Your valuable comments helped improve this manuscript.
We have revised the manuscript according to all your comments. Changes to the text
are highlighted in yellow in the revised manuscript.
Major:
1. Introduction/Discussion: Please elaborate on the clinical rational for this study.
Gene signatures are known to be prognostic. However, they do not guide clinical
decision making and especially do not change the selection of therapy.
Response 1: Thank you very much for pointing out that the clinical rationale for our
study remains unclear. We have added details regarding the aim of our study.
Revised manuscript: Abstract section (page 1, line 25-27), Introduction section (page
2, line 83-85) & Discussion section (page 10, line 293-294, page 11, line 298-299)
2. Can you provide data showing the correlation between immune cell status and
clinical risk stratification based on IMDC Risk Criteria? Most studies do not improve
the clinical standard of IMDC risk criteria in RCC...
Response 2: Whether immune cell status correlates with IMDC risk criteria, leading to
development in a clinical setting, is interesting. However, in this manuscript, we could
not apply IMDC risk criteria, which predicts survival for metastatic RCC patients. First,
we used a localized cohort (cT1-4N0-1M0), and the study endpoint is recurrence-free
survival. Second, of the 436 patients in the principal cohort, only 15 (3.4%) died of
ccRCC, suggesting that the analysis regarding cancer-specific survival yields
insufficient results. Regarding the TCGA cohort, we could not obtain information
regarding IMDC risk criteria. Therefore, we did not investigate the correlation between
immune cell status and IMDC risk criteria.
Revised manuscript: We have added the information on cancer-specific mortality in
the result section (page 5, line 188)
3. Table 2: Please add p-values
Response 3: Following your recommendation, I have added p-values in Table 2.
Because statistical significance was found between the two cohorts in each
methodology, we have added one sentence in the Discussion section.
Revised manuscript: Table 2, Results section (page 6, line 194-195), and Discussion
section (page 12, line 377-379)
Minor:
Paragraph 2 of the Introduction: Please improve the clinical rational to perform the
study. In addition to “technological and economical issues” the lack of a clear target
makes most approaches a failure. Therefore, most genomic applications have failed in
routine testing in urology/RCC despite being feasible (and even reimbursed in some
countries) (e. g. Rodler et. al., European Journal of Cancer 2021). Consequently, an
easy to perform histological assessment to predict response would be helpful. Please
discuss...
Response: Thank you very much for your suggestions. The clinical rationale of the
present study is to clarify whether the morphologic phenotype identified by H&E-
stained slides reflects clinical outcomes and the underlying genomes, applicable in a
routine clinical setting. Therefore, we have changed the sentence at the beginning of
Paragraph 2 in the Introduction section. We also added a recent reference regarding
the correlation between histologic immune status, using genetic analysis and multiplex
immunofluorescence/immunohistochemical staining [Au L et al, 2021 Cancer Cell].
Revised manuscript: Introduction section (page 2, line 56-60), and Reference [8]
Reviewer 2 Report
The present manuscript Histologic-based tumor-associated immune cells status in clear cell renal cell carcinoma correlates with gene signatures related to cancer immunity and clinical outcomes debates an actual and interesting topic, with impact on kidney cancer research. The title is clear and easy to be individualized in medical literature research. The abstract is well structured, but the aim of the study is not presented.
The Introduction is well documented, but the aim of the study is not clearly stated in the last paragraph of the chapter. I would include Figure 1 (study design) in Material and Method Section. In Introduction Section, I recommend a longer presentation of histological features of ccRCCC. Material and Methods are well conceived, to make the study reproducible.
The Results cover all the required fields. I would introduce as a supplementary file a table with groups clinical and biological characteristics.
Discussions are well conceived and support the Results. The manuscript includes few phrases about the study limitations but the authors do not clearly mention the impact of the study on the literature research.
The Conclusions do not strongly reflect the idea of the title. I recommend a larger presentation of this Section.The manuscript presents a recent bibliography, with a reasonable number of titles.
Author Response
Comments and Suggestions for Authors
The present manuscript Histologic-based tumor-associated immune cells status in clear
cell renal cell carcinoma correlates with gene signatures related to cancer immunity and
clinical outcomes debates an actual and interesting topic, with impact on kidney cancer
research.
We thank you for your valuable comments and appreciate the time and effort you have
taken to review our manuscript. We have revised our manuscript accordingly. Changes
to the text are highlighted in yellow in the revised manuscript.
The title is clear and easy to be individualized in medical literature research.
Response: Thank you for your confirmation. We have not revised the title.
The abstract is well structured, but the aim of the study is not presented.
Response: We added the aim of the study in the abstract. Correspondingly, we revised
the conclusion of the abstract. In addition, some words in the abstract have been deleted
in response to the character limit.
Revised manuscript: Abstract section (page 1, line 25-27, 38-40).
The Introduction is well documented, but the aim of the study is not clearly stated in the
last paragraph of the chapter.
Response: Following your recommendation, we added the study aim in the last
paragraph of the Introduction section.
Revised manuscript: Introduction section (page 2, line 83-85)
I would include Figure 1 (study design) in Material and Method Section.
Response: Following your suggestions, we have moved Figure 1 to the Materials and
Methods section.
Revised manuscript: Material and Methods section (page 3, line 110)
In Introduction Section, I recommend a longer presentation of histological features of
ccRCC.
Response: Following your recommendation, we added a description of histological
features of ccRCC with two additional references [12,13].
Revised Manuscript: Introduction section (page 2, line 70-72).
Material and Methods are well conceived, to make the study reproducible.
The Results cover all the required fields. I would introduce as a supplementary file a
table with groups clinical and biological characteristics.
Response: Thank you very much for pointing out that our manuscript did not include
tables of the clinicopathological characteristics associated with tumor-associated
immune cells status. We added Supplementary Tables S1-6.
Revised manuscript: Supplementary Tables S1-6, Result section (page 6, line 195-
197)
Discussions are well conceived and support the Results. The manuscript includes few
phrases about the study limitations but the authors do not clearly mention the impact of
the study on the literature research.
Response: We agree that we need to add another limitation to this study. As we used
the same cohort as our previous report, multiple institutional cohorts should be validated.
We added one sentence to the limitation section.
Revised manuscript: Discussion (page 12, line 369-371)
The Conclusions do not strongly reflect the idea of the title. I recommend a larger
presentation of this Section.
Response: Following your suggestions, we have revised the Conclusion section.
Revised manuscript: Conclusion section (page 12, line 385-388)
The manuscript presents a recent bibliography, with a reasonable number of titles.
Response: Thank you for your confirmation. We added three references accordingly.
Revised manuscript: Reference section ([8] [12] [13] have been add